# Molecular insights into the unusually promiscuous and catalytically versatile Fe(II)/α-ketoglutarate-dependent oxygenase SptF

Hui Tao[1,5], Takahiro Mori [1,2,3,5 ✉], Heping Chen[1], Shuang Lyu[1], Akihito Nonoyama[4], Shoukou Lee[4] & Ikuro Abe [1,2 ✉]

Non-heme iron and α-ketoglutarate-dependent (Fe/αKG) oxygenases catalyze various oxidative biotransformations. Due to their catalytic flexibility and high efficiency, Fe/αKG oxygenases have attracted keen attention for their application as biocatalysts. Here, we report the biochemical and structural characterizations of the unusually promiscuous and catalytically versatile Fe/αKG oxygenase SptF, involved in the biosynthesis of fungal meroterpenoid emervaridones. The in vitro analysis revealed that SptF catalyzes several continuous oxidation reactions, including hydroxylation, desaturation, epoxidation, and skeletal rearrangement. SptF exhibits extremely broad substrate specificity toward various meroterpenoids, and efficiently produced unique cyclopropane-ring-fused 5/3/5/5/6/6 and 5/3/6/6/6 scaffolds from terretonins. Moreover, SptF also hydroxylates steroids, including androsterone, testosterone, and progesterone, with different regiospecificities. Crystallographic and structure-based mutagenesis studies of SptF revealed the molecular basis of the enzyme reactions, and suggested that the malleability of the loop region contributes to the remarkable substrate promiscuity. SptF exhibits great potential as a promising biocatalyst for oxidation reactions.

[1] Graduate School of Pharmaceutical Sciences, The University of Tokyo, Tokyo, Japan. [2] Collaborative Research Institute for Innovative Microbiology, The University of Tokyo, Tokyo, Japan. [3] PRESTO, Japan Science and Technology Agency, Saitama, Japan. [4] Sumitomo Dainippon Pharma Co., Ltd, Osaka, Japan. [5] These authors contributed equally: Hui Tao, Takahiro Mori. ✉email: tmori@mol.f.u-tokyo.ac.jp; abei@mol.f.u-tokyo.ac.jp

O xygenases are widely distributed in nature and catalyze the incorporation of oxygen into various natural products. One of the largest oxygenase groups is the non-heme iron and α-ketoglutarate-dependent (Fe/αKG) oxygenase superfamily, which is involved in both primary and secondary metabolism[1]. In animals, Fe/αKG oxygenases play important roles in physiological processes via hydroxylation or *N*-demethylation reactions, with substrates including proteins, nucleic acids, and lipids[1]. In plants and microorganisms, in addition to the typical hydroxylation reactions, Fe/αKG oxygenases catalyze a wide range of chemical transformations in the biosynthesis of natural products[2–27], including complex skeletal rearrangement[11,12], ring-expansion[15], and C–C bond formation[21–23]. Given their intriguing chemical reactions, detailed biochemical, structural, and calculation studies of Fe/αKG oxygenases have been conducted over the past few decades[5,6,8,28–32].

Fe/αKG oxygenases utilize αKG as a co-substrate and Fe(II) as a cofactor[33–35]. Through the oxidative decarboxylation of αKG, a reactive Fe(IV)-oxo intermediate is generated to activate a selective aliphatic C–H bond, which is normally difficult to cleave due to high bond-dissociation energy, and then oxidative products are generated following radical recombination. Unlike other oxidases, including P450 monooxygenases, Fe/αKG oxygenases do not require any reductase partners or expensive cofactors such as NAD(P)H and FAD/FMN[36]. Furthermore, these enzymes exhibit high catalytic efficiencies, and typically have large turnover numbers[11,14,37,38]. These features of Fe/αKG oxygenases, along with their catalytic versatility, have attracted considerable attention as a source of biocatalysts[36,39].

SptF is an Fe/αKG oxygenase first discovered in the biosynthesis of the fungal meroterpenoid emervaridones[40]. It catalyzes two oxidation steps to form emervaridone C (**3**) from andiconin D (**1**) (Fig. 1, Supplementary Fig. 1). The proposed reaction scheme involves the initiation by a C11 hydroxylation reaction, followed by water elimination, carbon skeletal rearrangement, and deprotonation reactions to yield **2**. Compound **2** is then accepted by SptF again and undergoes an epoxidation reaction to generate **3**. Thus, SptF accepts two skeletally distinct molecules as substrates and plays a key role in the structural complexification of andiconin-derived meroterpenoids. Although the function of SptF has been investigated by in vitro enzymatic reactions, the detailed mechanisms of the dynamic skeletal rearrangement and the accommodation of structurally distinct substrates remain to be elucidated.

In this study, we investigated SptF further by in vitro enzyme reactions, crystallization, and structure-based mutagenesis. To our surprise, SptF is an unusually promiscuous and catalytically versatile enzyme that performs up to four rounds of oxidative reactions with its natural substrate **1**, including hydroxylation, desaturation, epoxidation, and skeletal rearrangements. SptF also catalyzed the formation of cyclopropane-ring-fused, highly congested 5/3/5/5/6 and 5/3/6/6/6 cyclic skeletons from structurally distinct meroterpenoid terretonins. Moreover, SptF hydroxylated steroids, including androsterone, testosterone, and progesterone, with different regiospecificities. Our crystallographic analyses of SptF in complex with (un)natural substrates, along with our structure-based mutagenesis studies, have revealed the molecular basis for the substrate recognition and suggested that the malleability of the loop region contributes to the remarkable substrate promiscuity.

## Results

**Substrate promiscuity and catalytic versatility.** Emervaridone C (**3**) was previously reported as the final product of SptF from andiconin D (**1**)[40]. However, our careful reexamination of the enzyme reaction revealed that SptF actually produced two more products **4** and **5**, in addition to **2**, **3**, and previously reported emeridone F[40] (Fig. 1a and Supplementary Fig. 1). Structural analyses established that **4** contains an additional C9 hydroxyl group and C1–C2 epoxide (Fig. 1c, Supplementary Figs. 2, 19-24 and Supplementary Table 8). In contrast, **5** has a new ether bridge between C4' and C2', along with a C5' hydroxyl group (Fig. 1c, Supplementary Figs. 25-30 and Supplementary Table 9). Notably, SptF, in the emervaridone biosynthetic pathway, shares 94% amino acid sequence identity with AndF, which is responsible for the generation of anditomin (**7**) from andilesin C (**6**), in the closely related anditomin pathway (Fig. 1c)[37]. As expected, SptF also accepted **6** as a substrate to produce **8** and **9** in addition to **7** (Fig. 1b, c, Supplementary Figs. 31-42 and Supplementary Tables 10 and 11). Interestingly, **8** and **9** share the same skeleton, but have different stereochemistry at the C3'–C9' epoxide ring. To determine the reaction sequence, putative intermediates, including **2**, **3**, and emeridone F were used as substrates of the SptF enzyme reactions. As a result, both **2** and **3** were efficiently converted into **4** and **5**, which clearly indicated that SptF converts **1** into **3** via **2**, and then **4** and **5** are generated from **3** (Supplementary Fig. 11c, d). Similarly, **8** and **9** were generated from **6** via **7** (Supplementary Fig. 11e). In contrast, emeridon F with the C9β hydroxyl group was not accepted by SptF, indicating that it is a shunt product. It is remarkable that SptF, a single enzyme, accepts structurally distinct meroterpenoids as substrates and catalyzes sequential oxidation reactions, including desaturation, epoxidation, hydroxylation, ring-opening, and carbon skeletal rearrangements, to generate skeletally diverse products.

To investigate the origin of oxygen atoms incorporated by SptF, we performed labeling experiments using $^{18}O_2$ and $H_2^{18}O$ (Supplementary Table 1 and Supplementary Fig. 3). When **1** was incubated with SptF using $^{18}O_2$ (98 atom$^{18}$O%), +2 of **3** ($m/z$ 428 $[M + NH_4]^+$) was observed ($m/z$ 426 (+0): 428 (+2) = 2.2%: 97.8%). On the other hand, in the experiment with $H_2^{18}O$ (final concentration 78 atom $^{18}$O%), small amount of $^{18}$O was incorporated into **3** ($m/z$ 426 (+0): 428 (+2) = 95.6%: 4.4%). Further, double labeling of $H_2^{18}O$ and $^{18}O_2$ revealed that the enzyme generated +2 of **3** ($m/z$ 428) as a major product together with +0 ($m/z$ 426) and +4 ($m/z$ 430) of **3** as minor products ($m/z$ 426 (+0): 428 (+2): 430 (+4) = 1.3%: 92.0%: 6.8%). These results indicated that an epoxide group introduced by SptF is mostly from $O_2$, while ~5% of water-derived oxygen atom are also incorporated into **3**. This would be caused by solvent exchange of ferryl and/or ferric species with water during the catalytic cycle[41] and/or nonenzymatic reaction of **3** with water molecule. Indeed, the incubation of **3** with 78% $H_2^{18}O$ without SptF revealed the incorporation of an oxygen atom from water ($m/z$ 426 (+0): 428 (+2) = 94.6%: 5.4%), indicating that an oxygen atom of **3** can be nonenzymatically exchanged with water. A possible explanation is addition of water to a carbonyl group to yield a geminal-diol. Its collapse back to the carbonyl could proceed with elimination of $^{16}O$ water, explaining the $^{18}O/^{16}O$ exchange. This reaction may happen at C4' carbonyl of **3**. This carbonyl carbon becomes less reactive in **4**, because the C9 hydroxyl group blocks it, and also less reactive in **5**, where it is turned into an ester carbonyl. In fact, the observed $^{18}O/^{16}O$ exchange was not so significant in **4** and **5** as it was in **3**.

On the other hand, major +6 peaks ($m/z$ 447) were observed for **4** ($m/z$ 443 (+2): 445 (+4): 447 (+6) = 1.1%: 8.6%: 90.3%) and **5** ($m/z$ 443 (+2): 445 (+4): 447 (+6) = 0.9%: 3.4%: 95.7%) when we incubated **1** and SptF with 98% $^{18}O_2$. Furthermore, the double labeling experiments with 78% $H_2^{18}O$ and 98% $^{18}O_2$ showed similar labeling pattern to 98% $^{18}O_2$-labeling experiment

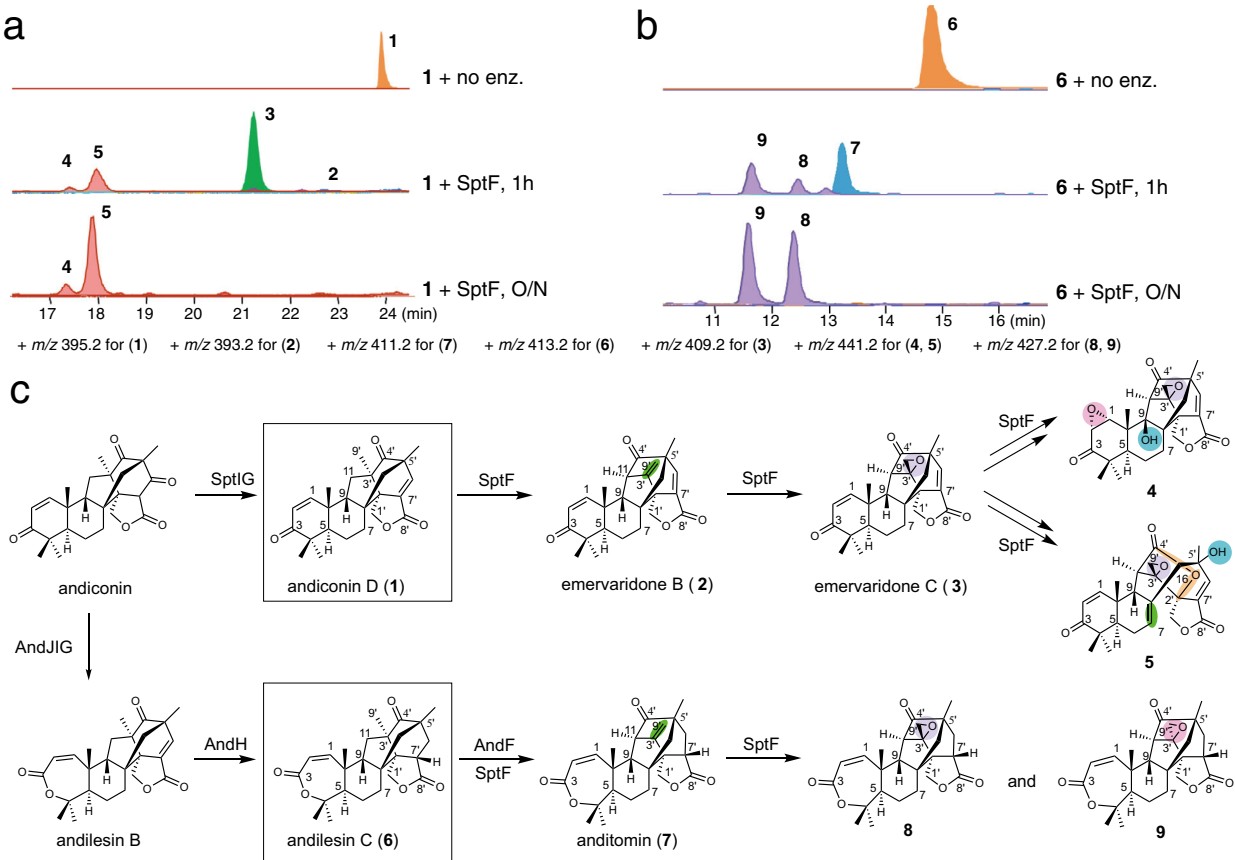

**Fig. 1 Characterization of SptF function in vitro using andiconin D (1) and andilesin C (6) as substrates. a** LC-MS EICs of products from the in vitro assay using **1** as the substrate, and **b** using **6** as the substrate (the +*m/z* value for each compound is shown). **c** Biosynthesis and structures of SptF and AndF products, using **1** and **6** as substrates in vitro. Note that the products **4**, **5**, **8**, and **9** were isolated from large scale enzymatic reactions, and their structures were determined by NMR.

(for **4** (*m/z* 443 (+2): 445 (+4): 447 (+6): 449 (+8) = 0.9%: 4.8%: 92.1%: 2.2%) and for **5** (*m/z* 443 (+2): 445 (+4): 447 (+6): 449 (+8) = 0.8%: 3.1%: 95.1%: 1.0%)). These results clearly indicated that all oxygen atoms incorporated into **4** and **5** are mostly derived from $O_2$. In contrast, the very small +8 peaks of **4** and **5** were observed in $H_2^{18}O$-labeling and double labeling experiments.

Encouraged by the promiscuity of SptF, we further explored its substrate scope by testing six fungal meroterpenoids with different core scaffolds, preandiloid B (**10**), preandiloid C (**12**), terretonin J (**13**), terretonin C (**15**), terretonin A (**17**), and terretonin (**19**) (Supplementary Fig. 4). Interestingly, SptF accepted all of them as substrates, except for **12**. SptF thus installed a hydroxyl group at C1 of **10** to form **11** (Fig. 2a, d, Supplementary Figs. 43-48, and Supplementary Table 12). Notably, SptF oxidized all four terretonins to yield products with 2 mass units less than the original (Fig. 2b-d), and structure determinations by spectroscopic analyses established that **14/18** and **16/20** possess highly congested, unique cyclopropane-ring-fused 5/3/6/6/6 and 5/3/5/5/6/6 scaffolds, respectively (Supplementary Figs. 2, 4, 5, 49-60, and Supplementary Tables 13, 14). The conversion rate of the enzyme reactions ranged from 6% to 51% over 24 h (Supplementary Table 2).

To further evaluate the utility of SptF as a biocatalyst, we also tested various steroids, including androsterone (**21**), testosterone (**24**), and progesterone (**28**), as substrates. To our surprise, SptF accepted **21**, **24**, **28** of them with high conversion rates of 34%-53% over 24 h (Supplementary Table 2). The structures of the products were determined by NMR and the crystalline-sponge-based X-ray

diffraction method (Fig. 3, Supplementary Figs. 6, 61-82, and Supplementary Tables 15-18). SptF thus accepted **21** and installed one hydroxyl group at C5 to yield **22**, and the second hydroxyl group at C9 to form the steroid **23**. In contrast, **24** underwent hydroxylation at C9 to yield **25**. Moreover, the hydroxylation and following oxidation at C6 of **24** generated **26**, which is interchangeable with **27** according to the NMR data. Although the structures of products **29–31** from **28** have not been determined yet, due to separation and stability problems, they are also predicted to be hydroxylated and/or oxidized products as judged from their molecular weights (Supplementary Fig. 2). The substrate promiscuity and catalytic versatility of SptF are thus quite exceptional. In contrast, ergosterol and lanosterol, produced by the natural host, were not accepted as a substrate by the enzyme.

**Substrate binding mode in the active site**. To elucidate the molecular basis of the promiscuous SptF enzyme reactions, we solved the structures of SptF-Fe/αKG/**1**, SptF-Fe/*N*-oxalylglycine (NOG; a catalytically inactive analogue of αKG[42])/**6**, and SptF S114A-Fe/NOG/**15** (Figs. 4a-c, 5a-c, Supplementary Fig. 7, and Supplementary Table 3). The structure of SptF shares the double stranded β-helix fold and conserved 2-His-1-Asp facial triad found in other fungal meroterpenoid Fe/αKG oxygenases, including AndA (PDB: 5ZM3)[12] with RMSD values of 1.4 Å for Cα-atoms and 38% amino acid sequence identities.

In the active site, **1** or **6** is located on the opposite side of the facial triad in an almost identical fashion (Fig. 4a, b and Fig. 5a, b), indicating that the size of the A-ring does not affect the binding mode. The distance between the iron and the initial

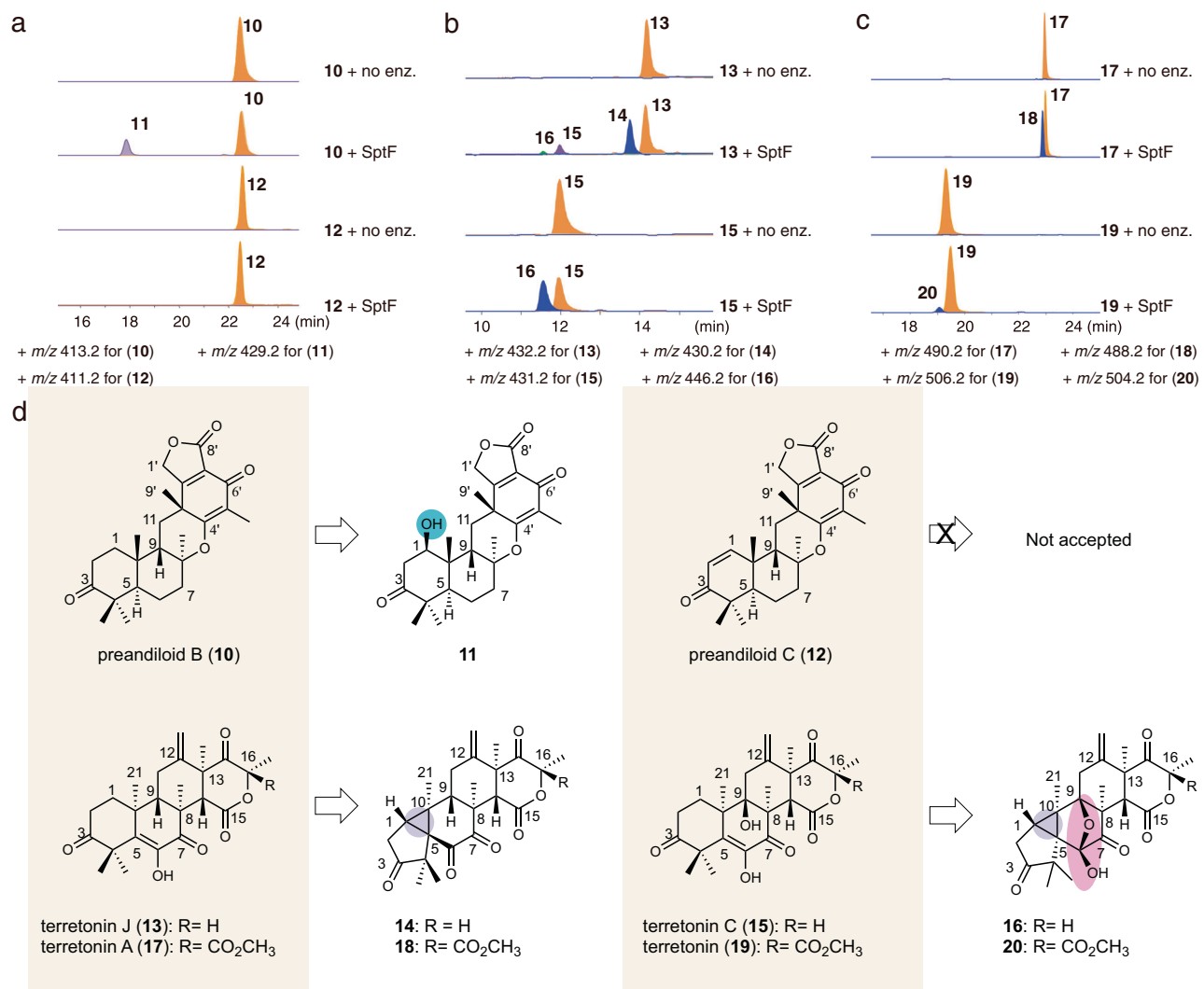

**Fig. 2 Substrate scope of SptF. a–c** LC-MS EICs for in vitro assays using preandiloid B (**10**), preandiloid C (**12**), terretonin J (**13**), terretonin C (**15**), terretonin A (**17**), and terretonin (**19**). The $+ m/z$ value used for each compound is shown. **d** Structures of substrates and corresponding products. Products **11**, **14**, and **16** were isolated from large scale enzymatic reactions, and their structures were determined by NMR. Note that the structures of **18** and **20** were deduced according to the HRMS, UV pattern, and reaction type (Supplementary Figs. 2 and 5).

reaction site C11 of **1** and **6** is 4.2 and 4.3 Å, respectively, which are close enough for hydrogen atom abstraction by the C–H bond activation. The active site hydrophobic residues, Ile63, Phe133, and Ile231, are located within van der Waals distances of the B/C-ring, suggesting that a hydrophobic surface facilitates the substrate binding. Similar to the previously reported AndA[12], the loop between Trp53 and Asn75 of SptF (Supplementary Fig. 8) was only observed in complex structures with **1** or **6** (Fig. 4a, b). This lid-like loop region interacts with the A-ring of **1** or **6** via only one hydrogen bond with Asn65, suggesting loose interactions between the lid-like region and substrate. In contrast, the D/E-rings of **1** and **6** are tightly fixed by several hydrophilic residues located on the bottom surface of the active site cavity: the carbonyl oxygen of the E-ring interacts with Ser114, while the carbonyl oxygen of the D-ring interacts with the main chains of Leu199 and Thr148 via water molecules (Fig. 4a, b).

Interestingly, the unnatural substrate **15** shows a different ligand binding mode in the active site. The D-ring of **15** forms a hydrogen-bond network with Thr148 and Leu199 via water molecules, but also forms a hydrogen bond directly with Asn150 (Figs. 4c and 5c). Notably, the lid-like loop region was not observed upon the binding of **15**, while the conformations of the

other active site residues were almost identical. In this structure, Phe133 and Ile231 are located close to the methyl groups of C8, C10, and C13, and potentially form hydrophobic interactions. The C6 enol group of **15** is close to the iron with a distance of 3.7 Å, suggesting that the enolic hydrogen atom is abstracted to initiate the cyclopropane-ring formation reaction via the proposed path c in Supplementary Fig. 9.

**Structure-based mutagenesis.** To investigate the importance of the active site residues, we performed structure-based mutagenesis. Firstly, the hydrophobic residues Ile63, Phe133, and Ile231, lining the active site cavity, were substituted with Ala (Fig. 6a-c, Supplementary Figs. 10-12). As a result, the I63A variant abolished the formation of **4** and **5**, and instead generated the new product **32** (Fig. 6d) in addition to **2**. The structure of **32** was determined to be an E-ring opened dicarboxylic acid, which is possibly formed by two rounds of oxidation at the C1' position of **1** (Fig. 6b, e, Supplementary Figs. 10, 83-88, and Supplementary Table 19). On the other hand, I63A oxidized **2** to **3**, and converted **3** to small amounts of **4** and **5** (Supplementary Fig. 11a, b), whereas F133A abolished the production of **4** and **5**, but accumulated the early-stage intermediates **2** and **3** (Fig. 6b). The

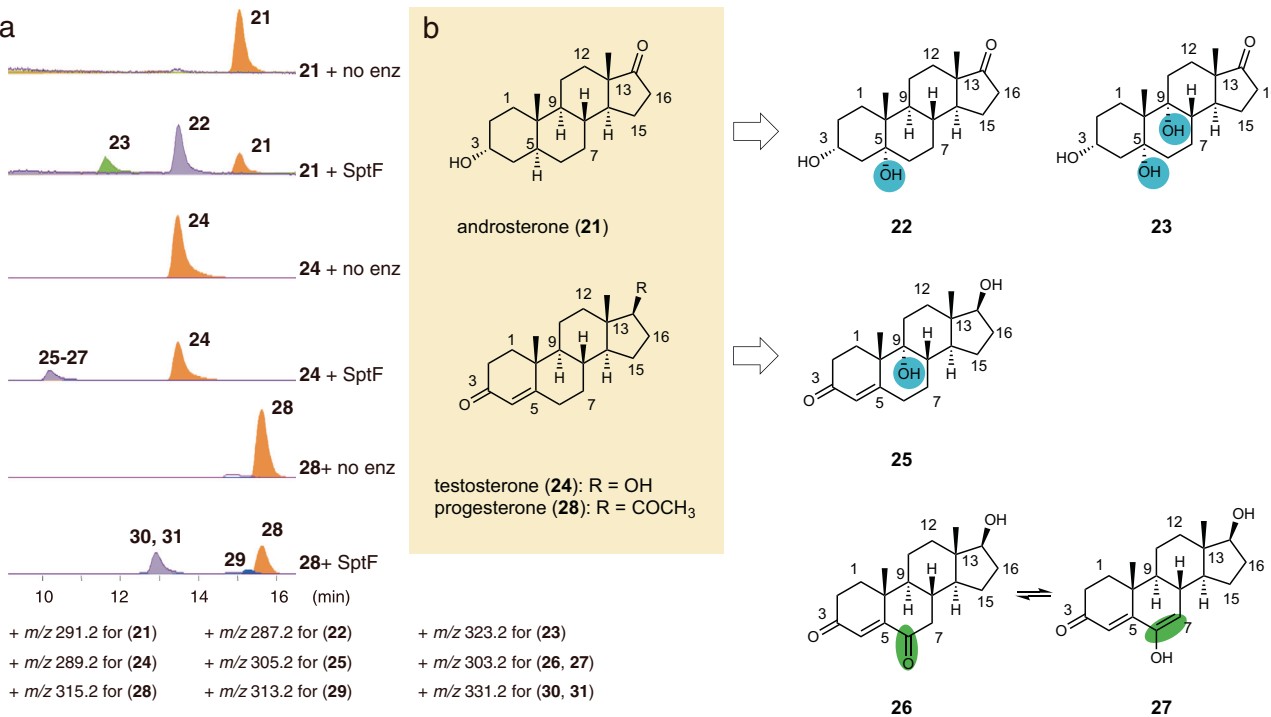

**Fig. 3 Oxidation of human steroid hormones by SptF. a** LC-MS EICs for in vitro assays using androsterone (**21**), testosterone (**24**), and progesterone (**28**). The + *m/z* value used for each compound is shown. **b** Structures of substrates and corresponding products. Products **22**, **23**, and **25**–**27** were isolated from large scale enzymatic reactions, and the structures of **22**, **23**, and **26**/**27** were determined by NMR, while the structure of **25** was solved by the crystal sponge method. Note that **26** and **27** are interchangeable with each other according to the NMR data. Products **29**–**31** were not determined due to separation problems, but their structures were deduced to be hydroxylated and/or oxidized compounds, considering their molecular weights (Supplementary Fig. 2).

$k_{cat}/K_M$ value of F133A for **1** decreased more than 1000-fold as compared with the wild-type enzyme (Supplementary Table 4). F133A also converted **1** into the C11-hydroxylated product **33** (Supplementary Figs. 89–94 and Supplementary Table 20). In contrast, the substitution of Phe133 with Tyr retained the generation of both **4** and **5** (Fig. 6b). Moreover, the I231A variant showed a similar LC-MS profile to that of F133Y, while affording **32** as the major product. The loss of the *sec*-butyl group increases the active site space around the D-ring of **1**, which could lead to an alternative substrate binding mode. Notably, similar results were observed in the enzyme reactions with the unnatural substrate **15**. The activity for the generation of the cyclopropane ring was abolished in F133A, but retained in F133Y. Moreover, I231A significantly reduced the activity to produce **16** (Fig. 6c), suggesting the similar role of the hydrophobic residues in the recognition of the unnatural substrate **15**. Compound **33**, generated by F133A variant, is a proposed biosynthetic intermediate of **2** (Supplementary Fig. 1)[40]. However, in vitro enzyme reaction of **33** with SptF revealed that **33** is not converted to the final products **3**–**5**, and therefore **33** is not an intermediate in the pathway (Supplementary Fig. 13).

Secondly, the hydrophilic residues that interact with substrates, including Asn65, Ser114, Thr148, and Asn150, were substituted with alanine (Fig. 6a-c). Due to the insolubility of T148A, the T148S variant was used for the enzyme reaction. As a result, N65A significantly reduced the **4** and **5** formation activity to 20% and accumulated **3** (Fig. 6b, e), suggesting that the hydrogen-bond interaction with Asn65 is not completely essential but moderately important for the binding of **3**. Interestingly, as compared with wild-type SptF, the activities for the formation of **4** and **5** were maintained at 83%, 93%, and 52% in the S114A, T148S, and N150A variants, respectively. The ratio of **4** and **5** was

altered to 7:3 in S114A and T148S, as compared with 4:6 in the wild-type, and decreased to 1:9 in N150A (Fig. 6e). Our observations suggested that these three residues determine the position of the hydrogen atom abstraction (C9 or C7) by anchoring **3** within the binding pocket, to generate **4** and **5** via a hydrogen-bond network.

Asn65 in SptF is uniquely substituted with threonine in the closely related AndF (Supplementary Fig. 12). To clarify the function of Asn65, we constructed the SptF N65T variant, which converted **1** into **5** as the major product, with a slight amount of **4** (Fig. 6b, e). Notably, in the crystal structure of SptF N65T-Fe/NOG/**1**, a hydrogen bond was observed between the C3 ketone at the A-ring and the newly introduced Thr65 (Figs. 5d, 6f, and Supplementary Table 3). These results indicated that a subtle change of the hydrogen-bond interactions affects the position of the hydrogen atom for abstraction and determines the product specificity. Interestingly, only the N150A variant lacked the activity for the conversion of **15** into **16** (Fig. 6b, c), suggesting that the direct hydrogen bonding between Asn150 and **15** is crucial for the substrate binding.

**Lid-like loop region and substrate recognition.** The mutagenesis studies revealed that the hydrogen-bond and hydrophobic interactions in the active site cavity are important for the substrate recognition. In contrast, while Ile63 and Asn65 on the lid-like loop region are important for the formation of **4** and **5**, these residues are not essential for the reaction with the unnatural substrate **15**, suggesting that **15** is mainly recognized by active site residues but not via residues on the lid-like loop. Indeed, the kinetic analysis of I63A and N65A variants toward **1** and **15** revealed that the $k_{cat}/K_M$ values for these variants towards **15** were comparable to that of wild-type SptF due to the decrease of

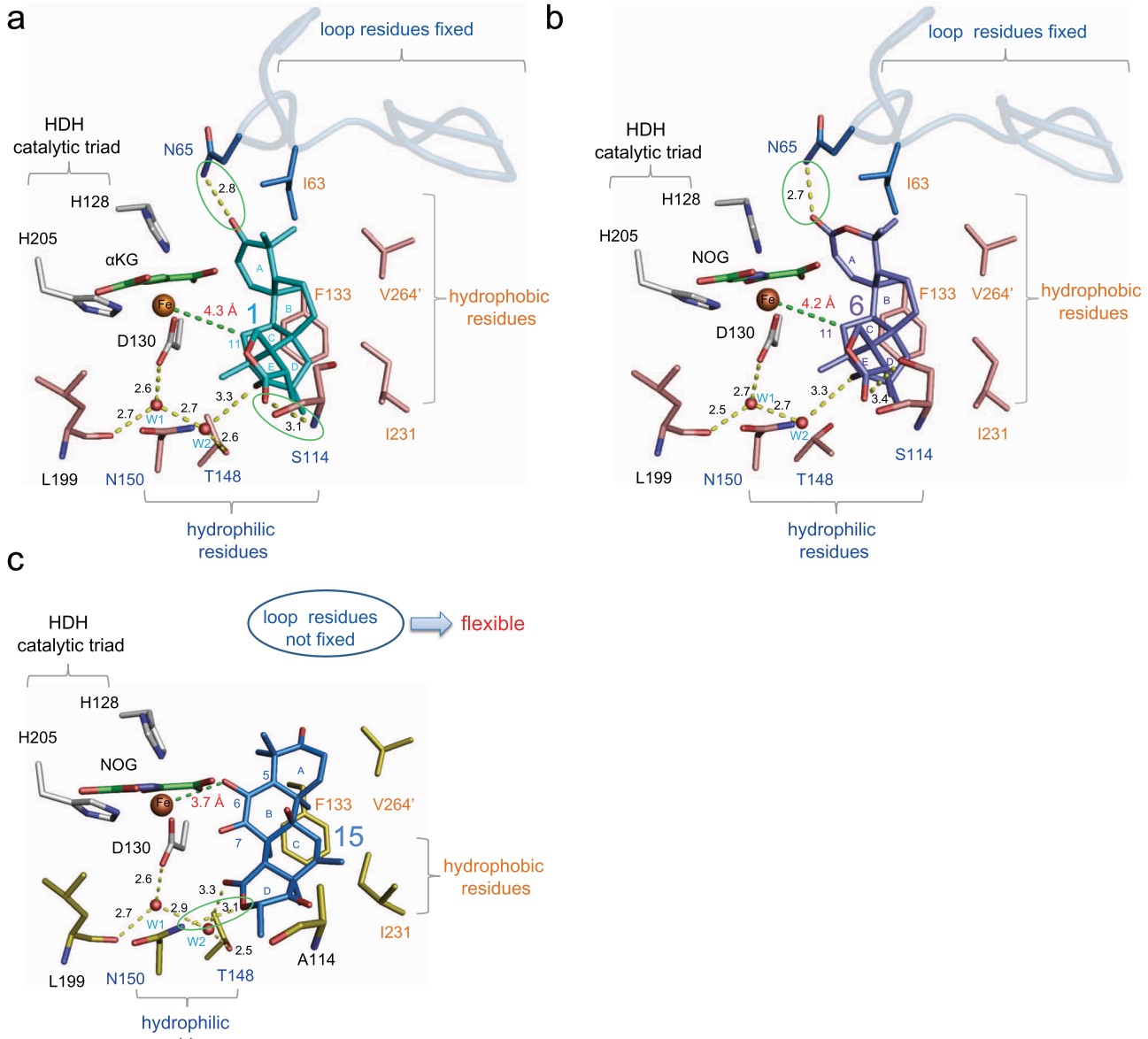

**Fig. 4 Active site view of SptF in complex with natural substrate or unnatural substrate. a** Active site residues interact with andiconin D (**1**), including hydrogen bonding with Asn65 and Ser114, indirect interactions with Thr148 and Leu199 via waters in SptF (shown in salmon). **b** Active site residues interact with andilesin C (**6**), including hydrogen bonding with Asn65 and Ser114, indirect interactions with Ler199 via waters in SptF. **c** Active site residues interact with terretonin C (**15**), including hydrogen bonding with Asn150, and indirect interactions with Thr148 and Leu199 via waters in S114A (shown in yellow). Note that the lid-like loop region, shown as blue cartoon, was only observed in the complex structure with **1** or **6**. Water molecules are shown as red spheres, and Fe are shown as orange spheres. αKG/NOG, **1**, **6**, and **15** are shown as green, cyan, purple, and blue sticks, respectively. Hydrogen networks are shown as yellow dash lines, and distances from Fe are shown with green dash lines. Key hydrogen-bond interactions are shown in green boxes.

$K_M$, while the $k_{cat}/K_M$ values of N65A for **1** was 14 times lower than that of wild-type. (Supplementary Table 4).

To further investigate the importance of the flexible lid-like loop region, with a length of 19 amino acid residues, we tested truncated loops. Based on the sequence comparison as well as the structures of SptF in complex with **1**, loop-truncated variants of 6 residues (His61-Val66), 9 residues (Lys58-Val66), 16 residues (Asn54-Lys69), and 19 residues (Trp53-Lys71) were constructed (Supplementary Figs. 14 and 15). Since the Δ19 variant was expressed in inclusion bodies, the Δ6, Δ9, and Δ16 variants were tested for their enzyme reaction activities with **1**, **2**, and **3**. Interestingly, although the activities of the truncated variants were dramatically decreased as compared to the wild-type, all the loop-truncated variants still consumed 60–94% and 17–77% of **1** and **2** over 24 h, respectively, and generated many minor products, including the E-ring-opened product **32**, but not **3–5** (Fig. 6b, Supplementary Fig. 16 and Supplementary Table 6). The differences in the activities among Δ6, Δ9, and Δ16 should be due to the folding of the enzymes. On the other hand, these variants also showed 26–56% activities toward **3** and generated small amounts of **4** and **5**, as well as other minor compounds (Supplementary Fig. 16 and Supplementary Table 6). Although we could not determine the structures of the minor products due to low yields, these results suggested that the lid-like region is not essential for the reactions with **1–3**, but important for the reaction efficiency and product selectivity.

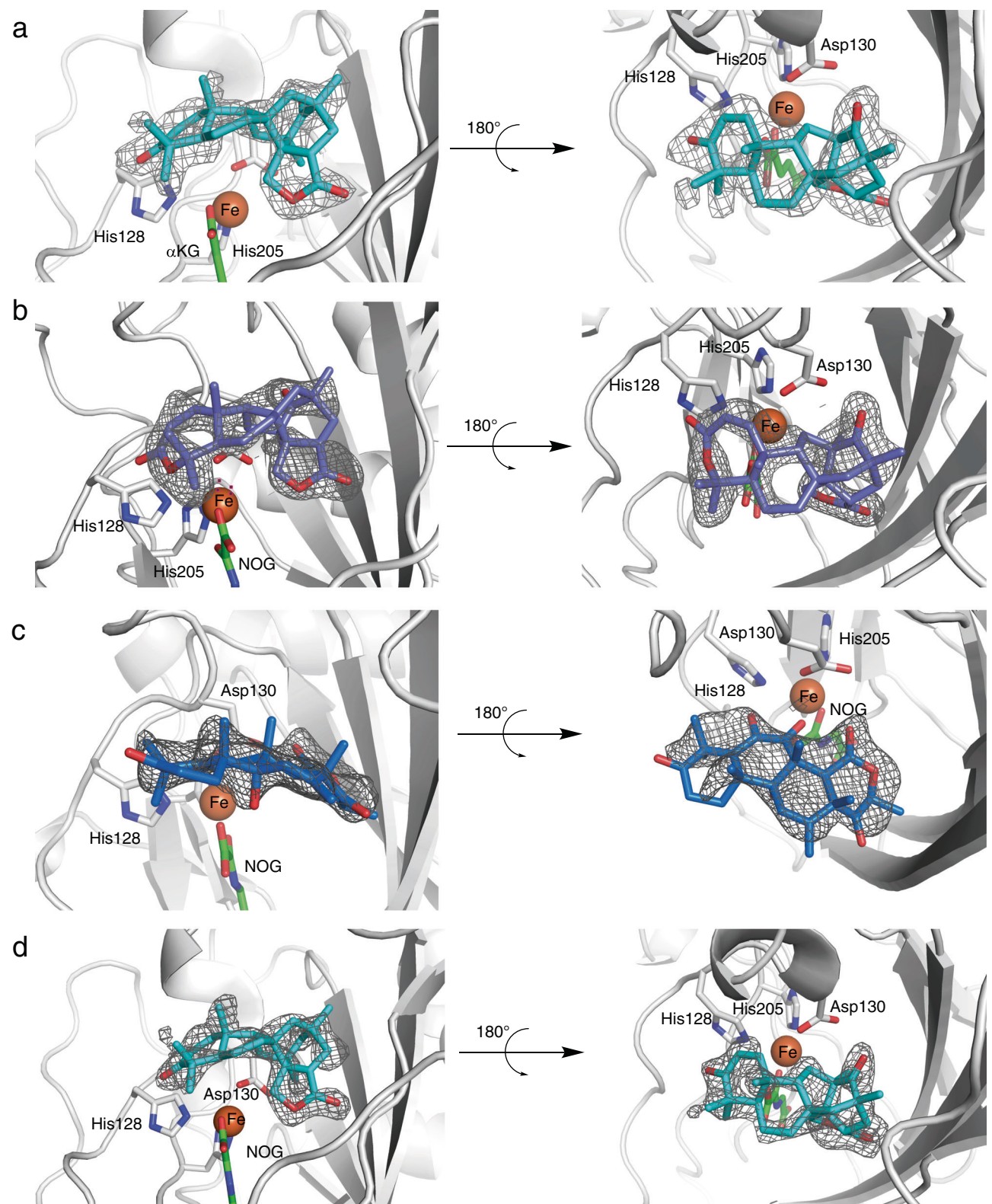

**Fig. 5 The omit map of the bound ligand in the SptF or mutant structures. a** SptF-Fe/αKG/1. **b** SptF-Fe/NOG/6. **c** SptF S114A-Fe/NOG/15. **d** SptF N65T-Fe/NOG/1. The Fo-Fc polder omit map of ligands are represented as a black mesh, contoured at +3.0 σ. Coloring scheme for ligands are the same as in Fig. 4.

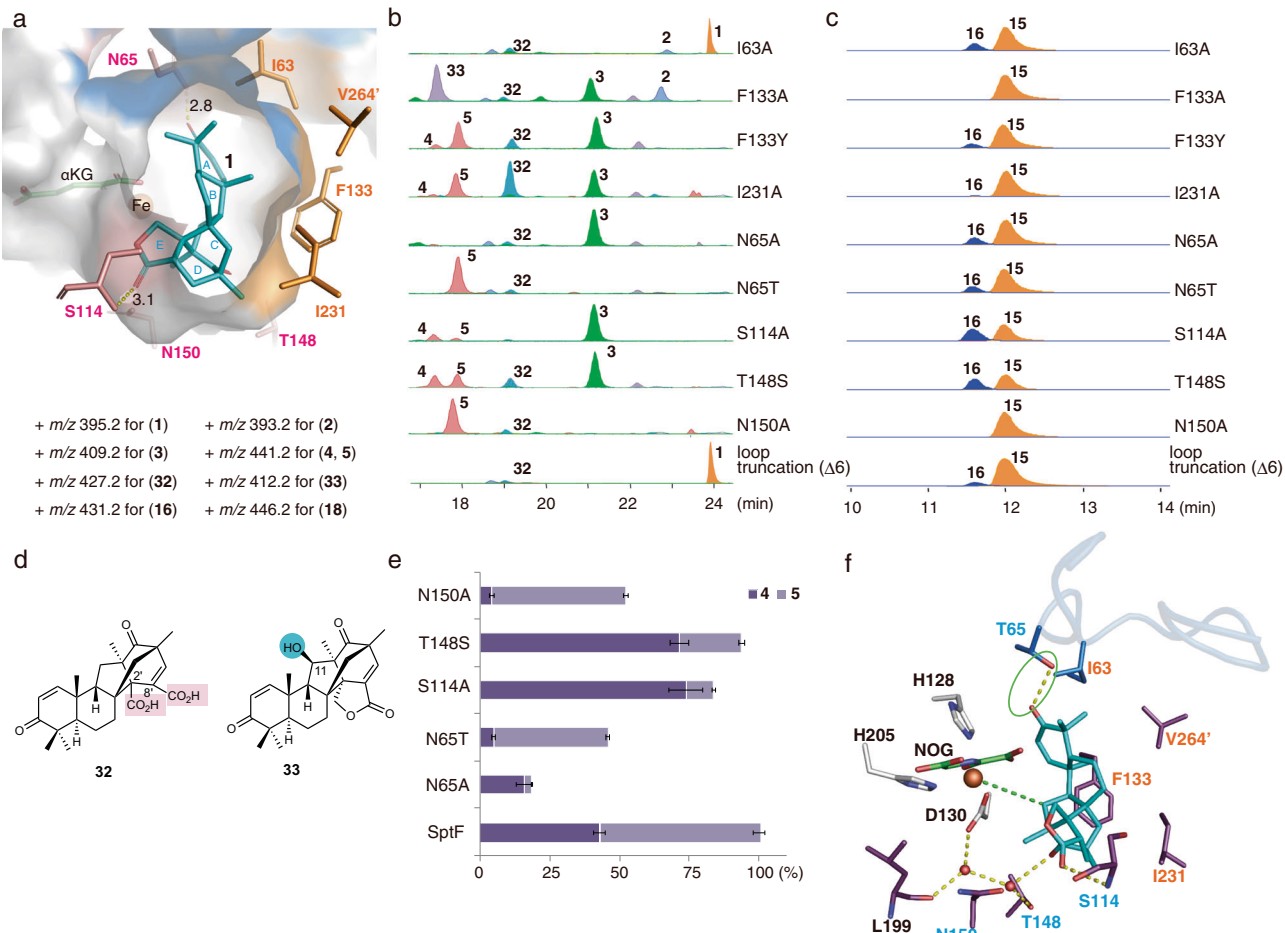

**Fig. 6 Mutagenesis studies. a** Active site cavity of SptF with **1** (shown as cyan sticks). Hydrophobic residues (shown as orange sticks), Ile63, Phe133, Ile231, and V264', form a hydrophobic surface to interact with the hydrophobic part of the substrate, whereas hydrophilic residues (shown as salmon sticks), Asn65, Ser114, Thr148, and Asn150, interact with the hydrophilic part of the substrate. **b**, **c** LC-MS EICs of products from in vitro assays with mutants using **1** or **15** as the substrate. The + $m/z$ value used for each compound is shown. **d** Structures of new products generated by mutants. Compounds **32** and **33** were isolated from large scale enzymatic reactions using the I231A and F133A mutants, respectively. Structures were determined by NMR. **e** Ratios of **4** and **5** from mutants compared with those from wild-type SptF. All reactions were performed in triplicates. Data are presented as mean values and the error bars indicate standard deviations (SD). **f** Active site residues interacting with **1** (shown as cyan sticks), including hydrogen bonding directly with Ser114 and the newly introduced Thr65, and indirectly with Thr148 and Leu199 via waters in N65T (shown in purple). Note that the coloring scheme for ligands is the same as in Fig. 4.

We also tested the activities of the truncated variants with unnatural substrates, including the fungal meroterpenoids **10** and **13**, and steroids **21**, **24**, and **28**. As a result, all the variants still maintained 24–40% and 56–84% activities toward **10** and **13**, but dramatically decreased activities toward **21**, **24**, and **28** (<20%) (Fig. 6c, Supplementary Fig. 17, and Supplementary Table 6). These observations indicated that the lid-like loop region is crucial for the steroid substrate recognition, but not essential for the meroterpenoid substrates, which is consistent with the absence of the electron density of the loop in the complex structure with **15**. The kinetics analysis of the loop-truncation variant Δ9 revealed that the variants also maintained comparable activity toward **15** to that of wild-type (Supplementary Table 4).

Thus, the mutagenesis and kinetic assay results strongly suggest that the malleability of the lid-like loop region contributes to the remarkable substrate promiscuity and catalytic versatility of SptF.

## Discussions

Substrate promiscuity plays a key role in both biomedical and industrial applications, as it provides a starting point for protein engineering and drug design, enzyme evolution with

different reaction/substrate specificities, and chemoenzymatic total synthesis[36,39,43–46]. Therefore, the identification and investigation of promiscuous enzymes enrich our understanding of how the enzymes obtain such a wide range of diverse functions, and will further guide the rational engineering of these enzymes as biocatalysts for the production of useful molecules with improved biological activities.

The Fe/αKG oxygenase SptF exhibits unusual promiscuity and catalytic versatility, and catalyzes four sequential oxidation reactions in the biosynthesis of fungal meroterpenoid emervaridones. SptF also catalyzes the formation of unique cyclopropane-ring-fused, highly congested 5/3/5/5/6/6 and 5/3/6/6/6 molecular scaffolds from the structurally distinct meroterpenoid terretonins. Moreover, SptF also hydroxylates steroids, including androsterone, testosterone, and progesterone. It is particularly remarkable that SptF accepts both 3-keto-$\Delta^4$- and 3-hydroxy-steroids with similar efficiencies to catalyze the hydroxylation at the α-face of different positions and generate a series of hydroxylated steroids.

Hydroxylation of steroids has been previously reported for cytochrome P450 enzymes[47], including the human CYP3A subfamily and the engineered bacterial P450 BM3, which catalyzes

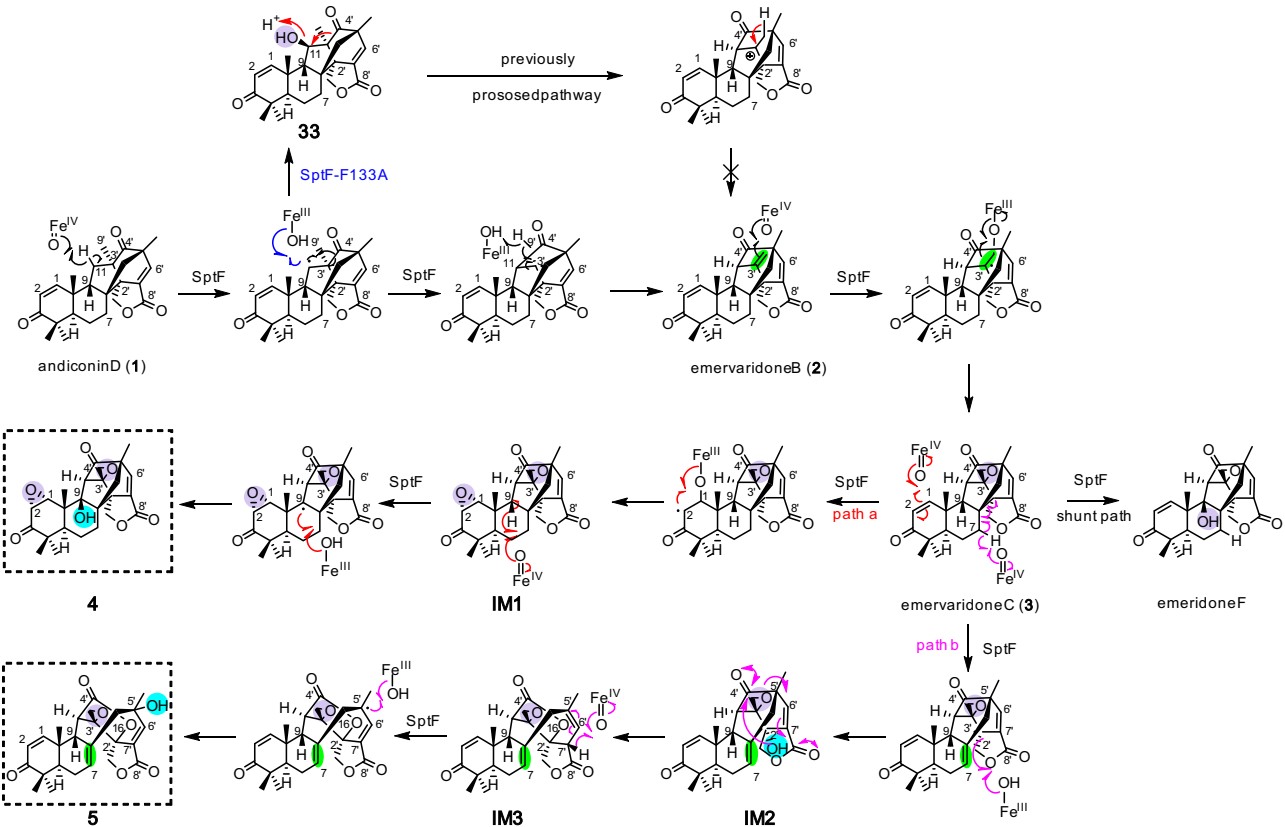

**Fig. 7 Proposed mechanism of SptF reactions.** The mechanism for the generation of emervaridone B (**2**) is revised in this study. Paths a (red) and b (magenta) branch from the intermediate **3**, to produce **4** and **5**, respectively.

the hydroxylation of testosterone, a 3-keto-Δ⁴-steroid, at the β-face of the C3 and C15 positions[48]. However, in spite of intensive engineering efforts, P450 BM3 does not accept 3-hydroxy-steroids[49]. Thus, SptF exhibits great potential as a promising biocatalyst for the oxidation of various natural products with important pharmacological activities.

Based on our structure-function analyses, we propose enzyme reaction mechanisms for the consecutive oxidation reactions in the emervaridone biosynthetic pathways, as follows (Fig. 7). Hydrogen abstraction at C11 of **1** by the ferryl species first generates a radical, which undergoes recombination to cleave the C–C bond between C4' and C3' and bridge C4' and C11, but not through the hydroxylated intermediate. Subsequently, a double bond is formed to yield **2**, which undergoes epoxidation to generate **3**. Compound **3** then serves as the branch point intermediate to produce **4** and **5**. In path a, epoxidation at A-ring of **3** generates **IM1**, which then undergoes hydroxylation at C9 to generate **4**. It should be noted that the C9 hydroxylation occurs with attack from the front side of the molecule, whereas all the other attacks by SptF can be explained from the back side, which would suggest that SptF can accept substrates with different binding mode. In path b, hydrogen abstraction at C7 induces the radical recombination, which cleaves the C8–C2' bond to construct a double bond between C7 and C8 on the B-ring. The hydroxyl rebound at C2' forms **IM2**, which is followed by lactone formation and C4'–C5' bond cleavage to yield **IM3**, in a similar manner to the formation of emeridone A[40]. In the last step, hydrogen abstraction at C7' and relocation of the radical to C5' complete the formation of the final product **5**. The LC-MS analysis of the enzyme reaction mixture suggested the presence of the intermediates **IM1-3**; however, their structures could not be determined because of instability and low yield (Supplementary Fig. 11).

The multifunctional catalytic activities largely stem from the unusual substrate promiscuity of SptF, which can accommodate a series of structurally different molecules in the active site for further oxidation reactions. The crystallographic and mutagenesis investigations have provided the structural basis for the promiscuity of SptF. In particular, SptF is unique since the lid-like loop region interacts with the natural substrate **1** via only one hydrogen bond with Asn65, which is clearly distinct from other fungal meroterpenoid Fe/αKG oxygenases, including AndA and PrhA[11,12]. In AndA and PrhA, the residues on the lid-like loop tightly recognize the D/E-rings of substrates via a hydrogen-bond network, while the A/B-rings loosely interact with the active site (Supplementary Fig. 18). These relaxed interactions and high malleability of the lid-like loop region in SptF are thought to contribute to its remarkable substrate promiscuity.

Truncations of the lid-like loop region demonstrated that the loop is not essential for the activity toward the natural and unnatural meroterpenoid substrates, but important for binding the steroid substrates and the product selectivity from natural substrates. The substrate is thought to be mainly recognized by the active site residues deep inside the cavity, which is then covered by the lid-like loop to further support the substrate loading and ensure the precise and efficient positioning for the C–H bond activation by the ferryl species. This whole process would be rigidly controlled for the natural substrate. In contrast, the flexible loop may interact loosely with the unnatural substrates, but still retains the efficiency and product specificity of the enzyme reactions. In particular, it would be difficult to tightly fix smaller steroids with a few functional groups in the active site. These molecules would probably loosely bind to the enzyme in several different conformations, and the enzymatic C–H bond activation could only take place when the substrate is properly

positioned for the interactions with the lid-like loop region. This would be the reason why we could not obtain the steroid-bound complex structure, despite our soaking experiments. Therefore, understanding how the lid-like loop region interacts with different substrates would provide key information to manipulate the SptF enzyme reaction.

In conclusion, we characterized the exceptionally promiscuous and catalytically versatile Fe/αKG oxygenase SptF, which catalyzes a remarkable repertoire of oxidation reactions of meroterpenoid and steroid molecules. We believe these unique features of SptF will serve as a platform for future efforts to optimize desired specific catalytic functions through directed evolution of the enzyme. SptF thus exhibits great potential as a promising biocatalyst for oxidation reactions.

## Methods

**General**. Oligonucleotide primers, listed in Supplementary Table 7, and DNA sequencing were ordered from Eurofins Genomics. The restriction enzymes *Nde*I and *Hin*dIII, and PrimeSTAR GXL DNA polymerase were purchased from Takara Bio Inc. Solvents and chemicals were purchased from Wako Chemicals, Ltd. (Tokyo, Japan), Merck KGaA Ltd. (Darmstadt, Germany), and Hampton Research (CA, USA), unless noted otherwise. PCR was performed using a TaKaRa PCR Thermal Cycler Dice® Gradient (Takara Bio Inc.). The NMR spectra of compounds were recorded on Bruker AVANCE III HD 800 MHz, AVANCE III HD 900 MHz (Bruker MA, USA), ECX-500 MHz, and EXZ-500 MHz (JEOL Ltd., Tokyo, Japan) spectrometers.

**Construction of SptF expression plasmid**. To obtain stable and pure proteins for in vitro enzyme reactions and crystallization experiments, expression plasmids for SptF lacking three residues, predicted to be flexible by XtalPred server[50], were prepared as follows. The primer pair SptF-N3-F/SptF-N3-R was used for amplification of the SptF DNA fragment from the previously reported pET-28a(+)-sptF[40] plasmid (see Supplementary Table 7 for primer sequence information). SptF PCR products were purified and ligated into the *Nde*I/*Hin*dIII-digested pET-22b (+) vector, using an In-Fusion® HD Cloning Kit (TaKaRa Clontech). The identities of the resulting vectors were confirmed by DNA sequencing. *Escherichia coli* DH5α was used for plasmid construction, and *E. coli* Rosetta™2(DE3)pLysS (Novagen) was used for the preparation of recombinant proteins.

**Construction of plasmids for expression of SptF mutants**. The primers used for the construction of plasmids for site-directed mutagenesis studies are listed in Supplementary Table 7. The plasmid for the expression of wild-type SptF was used as the template for PCR-based site-directed mutagenesis, which was performed with a QuikChange Site-Directed Mutagenesis Kit (Stratagene) according to the manufacturer's protocol. For the construction of loop-truncation mutants, the primer pairs listed in Supplementary Table 7 were used for the amplification of SptF DNA fragments from pET-28a(+)-sptF[40] plasmids. SptF PCR products were purified and ligated into the *Nde*I/*Hin*dIII-digested pET-22b (+) vector, using an In-Fusion® HD Cloning Kit (TaKaRa Clontech). The identities of the resulting plasmids were confirmed by DNA sequencing.

**Production and purification of SptF and its mutants**. In this study, the recombinant proteins used for in vitro assays were prepared as follows: Plasmids expressing SptF or its mutants were transformed into *E. coli* Rosetta™2(DE3) pLysS (Novagen). The resulting strains were cultured in LB medium supplemented with 34 mg/L chloramphenicol and 50 mg/L ampicillin sodium, with shaking at 37 °C. When the $OD_{600}$ reached 0.6, the cell cultures were cooled down on ice for 30 min, and then IPTG (0.2 mM) was added to induce the target protein expression at 16 °C. After 18 h of post-induction incubation, cells were harvested by centrifugation at $3300 \times g$ for 15 min and suspended in lysis buffer, containing 50 mM Tris-HCl (pH 8.0), 300 mM NaCl, 10 mM imidazole, and 5% glycerol. The cell suspension was sonicated for 4 min on ice. After removal of the cell debris by centrifugation at $20,400 \times g$ for 30 min, the supernatant was mixed with 1 mL Ni-NTA agarose resin and loaded onto a gravity flow column. Unbound proteins were removed with 50 mL lysis buffer containing 25 mM imidazole, and then the His-tagged protein was eluted with lysis buffer containing 300 mM imidazole. For the in vitro assay, the protein eluate was concentrated to 10 mg/mL after the removal of imidazole, using a 30 kDa Amicon® Ultra-15 filtration unit (Millipore). For crystallization, the eluate from the Ni-NTA agarose was applied to a HiLoad 16/60 Superdex 200 prepacked gel filtration column (4 °C, GE Healthcare), and eluted with a solution containing 20 mM Tris-HCl (pH 8.0), 100 mM NaCl, and 1 mM dithiothreitol. The resulting eluate was concentrated to 10 mg/mL, using an Amicon Ultra-4 (MWCO: 30 kDa) filter at 4 °C. The purity of the purified proteins was monitored by SDS-PAGE (Supplementary Fig. 15), and the protein concentrations were determined with a SimpliNano microvolume spectrophotometer.

**In vitro assay of SptF and its mutants**. Substrates used for in vitro enzyme reactions, including **1**, **6**, **10**, **12**, **13**, **15**, **17**, and **19**, were prepared according to previous studies[12,37,40,51], ergosterol and lanosterol were purchased from TCI (Japan). and the proteins were prepared as described above. All reactions in this study were performed following the previous studies unless noted otherwise[40]. Reactions were performed on a 50 μL scale with 50 mM PIPES (pH 7.5), 0.2 mM FeSO₄, 5 mM α-ketoglutarate, 4 mM ascorbate, 100 μM of substrate, and 15 μM of wild-type or mutant SptF. The reaction mixture without protein was used as the negative control. Enzyme reactions, except for steady-state enzyme kinetics, were incubated at 30 °C for 1 h or overnight and quenched by adding 50 μL methanol. After centrifugation and filtration, the reaction mixtures were analyzed by LC-MS. For comparisons of the substrate consumption between truncation variants and wild-type, the reactions were performed on a 50 μL scale with 50 mM PIPES (pH 7.5), 0.2 mM FeSO₄, 5 mM α-ketoglutarate, 4 mM ascorbate, 100 μM of substrate, and 40 μM of enzyme.

**Labeling experiments**. The labeling experiments were performed using previously reported methods[52]. Reactions were performed under anaerobic conditions in 500 μL tubes, in which 25 μL solutions, containing 5 μM SptF enzyme, 200 μM **1**, 5 mM αKG, 4 mM ascorbate, 200 μM FeSO₄, and 40 mM PIPES buffer (pH 7.5). Afterward, ¹⁸O₂ gas (98%) or air was injected into the vial. For the H₂¹⁸O labeling experiment, 20 mL of H₂¹⁸O (97%) was added for the preparation of 25 μL reaction solutions (final concentration of H₂¹⁸O is 78%). The enzyme reactions were performed at 20 °C for 10 min and quenched by adding equal volume of methanol. The prepared sample was stored at −80 °C until LC-MS analysis to decrease the exchange rate with solvents. The products were analyzed with the same method as that for the standard reaction products.

**LC-MS methods used for analyzing in vitro reactions**. All samples from in vitro assays were analyzed by a Bruker Compact QqTOF mass spectrometer (Bruker) coupled with an LC-20AD ultra-high-performance liquid chromatography (UHPLC) system (Shimadzu). MS data were recorded in the positive electrospray ionization mode. Compounds were separated on a COSMOSIL 5C18-MS-II Packed Column (2.0 × 100 mm, 5 μm). All reaction mixtures were separated using H₂O supplemented with 0.1% v/v formic acid as solvent A and acetonitrile as solvent B. The gradient for the separation of compounds in reaction mixtures with **1**, **10**, **12**, **17**, or **19** was 20% to 45% B over 20 min, then increase to 100% B in 1 min, and hold for 4 min at 40 °C with a flow rate of 0.2 mL/min. Reaction mixtures with **6** are shown here: 20–60% B over 17 min, then increase to 100% B in 0.1 min, and hold for 2.9 min at 40 °C with a flow rate of 0.15 mL/min. Reaction mixtures with **13** or **15** are shown here: 20–100% B over 20 min, and hold for 5 min at 40 °C with a flow rate of 0.15 mL/min. Reaction mixtures with **21**, **24**, or **28** are as follows: 10–100% B over 20 min, and hold for 5 min at 35 °C with a flow rate of 0.15 mL/min.

**Crystallization and structure determination**. Crystals of SptF lacking three residues, as well as the N65T and S114A mutants, were obtained after 1 day at 10 °C by using the sitting-drop vapor-diffusion method with the following reservoir solutions: SptF-apo: 0.27 M magnesium chloride, 21% w/v PEG 3350, 1.5% trimethylamine N-oxide dihydrate. SptF N65T: 0.2 M magnesium formate dehydrate, 20% w/v PEG 3350. SptF S114A: 0.27 M magnesium chloride, 20% w/v PEG 3350, 0.2 M sodium chloride, 1.5% trimethylamine N-oxide dihydrate. The protein concentrations used for the crystallizations of SptF, N65T, and S114A were 7.5 mg/L, 10 mg/L, and 7.5 mg/L, respectively. Since S114A can produce **4** and **5**, and has better diffraction as compared with wild-type SptF, we employed it for the soaking experiments of **15**. The complex structures were prepared by incubating SptF, N65T, or S114A crystals at 10 °C for 12 h with 5 mM of compound, 5 mM of αKG/NOG, and 100 μM of Fe in the crystallization drop. When αKG was used, the soaking experiment was performed in an anaerobic chamber.

The crystals were transferred into the cryoprotectant solution (reservoir solution with 25% (v/v) glycerol), and then flash cooled at −173 °C in a nitrogen-gas stream. The X-ray diffraction datasets were collected at BL-1A (Photon Factory, Tsukuba, Japan), using a beam wavelength of 1.1 Å;. The diffraction datasets for SptF were processed and scaled using the XDS program package[53] and Aimless in ccp4[54]. The initial phase of the SptF structure was determined by molecular replacement, using AndA (PDB ID: 5ZM4) as the search model. Molecular replacement was performed with Phaser in PHENIX[55,56]. The initial phase was further calculated with AutoBuild in PHENIX[57]. The SptF structures were modified manually with Coot[58] and refined with PHENIX.refine[59]. The cif parameters of the ligands for the energy minimization calculations were obtained by using the PRODRG server[60]. The ligands were added in the unassigned density at active site with several conformations and compared their 2mFo-DFc and mFo-DFc maps and Real-space fit (R-value, RSR, and corr. coeff., RSCC) values calculated by PHENIX.refine and PDB validation server, respectively (Supplementary Fig. 7). The final crystal data and intensity statistics are summarized in Supplementary Table 3.

The Ramachandran statistics are as follows: 98.3% favored, 1.7% allowed for SptF-apo, 97.7% favored, 2.3% allowed for SptF complexed with **1** and αKG, 98.5% favored, 1.5% allowed for SptF complexed with **6** and NOG, 97.7% favored, 2.3% allowed for SptF N65T complexed with **1**, 98.3% favored, 1.7% allowed for SptF

S114A complexed with **15** and NOG, and 98.5% favored, 1.5% allowed for SptF Δ9 complexed with NOG. All crystallographic figures were prepared with PyMOL (DeLano Scientific, http://www.pymol.org).

**Steady-state enzyme kinetics.** Each assay was performed in a total volume of 50 μL, containing 50 mM PIPES buffer (pH 7.5), 4 mM ascorbate, 200 μM FeSO₄, and 1 mM αKG, at 30 °C, and quenched by the addition of 50 μl methanol. Compounds **1** (0.5–25.0 μM), **6** (0.5–20.0 μM), and **15** (10.0–300.0 μM) were used to incubate with 0.025, 0.01, and 5.0 μM SptF for 3 min, 5 min, and 2 h to determine the kinetic parameters, respectively. Similarly, compounds **1** (1.0–100.0 μM) and **15** (10.0–200.0 μM) were used to incubate with 0.5 and 5.0 μM N65A for 5 min and 2 h to determine the kinetic parameters, respectively. For F133A, compounds **1** (20.0–300.0 μM) and **6** (10.0–200.0 μM) were used to incubate with 5.0 and 2.5 μM enzyme for 30 min, respectively. For I63A or loop-truncation Δ9 mutants, 5.0 μM enzyme was used to incubate with **15** (10.0–300.0 μM) for 2 h. Detailed concentration of substrates used to determine the kinetics of each enzyme were shown in the plot of Supplementary Table 4. All reactions were analyzed by LC-MS, and substrate consumption or product formation was determined by calculating the total peak areas of the substrate or product in each reactions as compared with negative control (without enzyme). For compound **1** and **6**, kinetic parameters were determined based on consumption of substrates, while for **15**, formation of product was used instead. Each reaction was performed in triplicate. GraphPad Prism (GraphPad Prism Software Inc., San Diego, CA) was used for statistics data analysis (Supplementary Table 4).

**Enzyme reaction products from 1, 6, 10, 13, and 15.** To isolate the products from **1**, **6**, **10**, **13**, and **15** for structure determination, in vitro enzymatic reactions were performed as described above in a total volume of 100 mL, divided into 0.5 mL portions in 2 mL tubes. After an overnight incubation at 30 °C, the reactions were quenched and extracted twice with an equal volume of ethyl acetate. After the ethyl acetate removal, 1 mL methanol was added to dissolve the products for separation by semi-preparative HPLC, which was performed on a Shimadzu LC20-AD HPLC system.

Compounds **4**, **5**, and **33** were purified from large scale enzymatic reactions as mentioned above, using wild-type SptF and the F133A mutant with **1**. Compounds **8** and **9** were prepared using wild-type SptF from **6**. These products were purified on an XSelect HSS T3 OBD Prep column (10.0 × 250 mm, 5 μm) using 50% methanol supplemented with 0.1% v/v formic acid at 40 °C. The injection volume was 0.1 mL and the flow rate was 2.5 mL/min. Finally, 0.3–0.4 mg of compounds **4**, **5**, **8**, **9**, and **33** were obtained.

Compound **32** was purified from large scale enzymatic reactions using I231A, on an XSelect HSS T3 OBD Prep column (10.0 × 250 mm, 5 μm) with H₂O supplemented with 0.1% v/v formic acid as solvent A and methanol as solvent B, at 40 °C. The gradient for separation was 46% B for 40 min, and 56% B for 60 min. The injection volume was 0.1 mL and the flow rate was 2.5 mL/min. The purification yielded 0.3 mg of compound **32**.

Compound **11** was purified with an XSelect HSS T3 OBD Prep column (10.0 × 250 mm, 5 μm), using 45% acetonitrile supplemented with 0.1% v/v formic acid, at 40 °C. The injection volume was 0.15 mL and the flow rate was 3.0 mL/min, and 0.9 mg of compound **11** was obtained.

Compound **11** was purified with an XSelect HSS T3 OBD Prep column (10.0 × 250 mm, 5 μm), using 45% acetonitrile supplemented with 0.1% v/v formic acid, at 40 °C. The injection volume was 0.15 mL and the flow rate was 3.0 mL/min, and 0.9 mg of compound **11** was obtained.

Compound **16** was purified with a COSMOSIL πNAP Packed column (10.0 × 250 mm, 5 μm), using 50% acetonitrile supplemented with 0.1% v/v formic acid, at 40 °C. The injection volume was 0.15 mL and the flow rate was 3.0 mL/min, and 0.9 mg compound **16** was obtained.

**Enzyme reaction products from 21 and 24.** Reactions were performed in a total volume of 100 mL, with 100 mM HEPES (pH 7.5), 0.2 mM FeSO₄, 4 mM α-ketoglutarate, 2 mM ascorbate, 1.6 mM of substrate, 10% glycerol, and 50 μM of SptF, at 4 °C for 56 h. Products were separated by normal-phase silica gel chromatography. The elution conditions were as follows: CHCl₃: MeOH = 100: 0.5 for 540 mL; then CHCl₃: MeOH = 100: 1 for 90 mL, and CHCl₃: MeOH = 100: 1.5 for 90 mL. Finally, 2.0 mg and 1.3 mg of compounds **22** and **23** were obtained from **21**, respectively. From **24**, 2.2 mg of **25** and **26/27** were obtained, respectively.

**Structure determination of 25 by crystalline-sponge method.** A single crystal of [(ZnCl₂)₃(tpt)₂(cyclohexane)ₓ] (tpt = 2,4,6-tris-(4-pyridyl)-1,3,5-triazine)[61] was used as the crystalline sponge in this study. The crystalline sponge (typically ~200 × 150 × 50 μm³) was placed in a vial and immersed in a small amount of cyclohexane (50 μL). A 1,2-dichloroethane solution (5 μL) of the compound (1 μg/μL, 5 μg) was added to the vial. The vials were equipped with a syringe needle for slow solvent evaporation. Guest soaking was carried out at 50 °C for 1 d. After confirming solvent evaporation, the X-ray diffraction datasets were collected at BL02B1 (SPring-8, Hyogo, Japan), using a beam wavelength of 0.4115 Å. The sample crystal was cooled to 100 K, using a cold nitrogen stream. The collected data

were integrated, corrected, and scaled using the program CrysAlisPro, ver. 1.171.38.46 (Rigaku Oxford Diffraction, 2015). All crystal structures were solved using the program SHELXT ver. 2014/5[62] and refined with the program SHELXL ver. 2018/1[63]. The crystal structure of the **25**/crystalline-sponge complex was solved and three independent molecules of **25** per asymmetric unit were clearly observed. All nonhydrogen atoms were refined anisotropically. All hydrogen atoms were generated using the proper HFIX command and refined isotropically using the riding model. All nonhydrogen atoms in the framework were refined without restraints and constraints. Guest molecules were refined with restraints on the atomic displacement parameters (SIMU and RIGU). Three independent guest molecules were characterized as geometrically equal guest molecules by restraint (SAME). The populations of the three independent guest molecules per asymmetric unit were estimated to be 69%, 56%, and 31%, according to the structure refinement. The solvent cyclohexane molecules suggested in the electron density map were refined using some restraints (DFIX, DANG, SIMU, and ISOR), because the corresponding electron densities were obscured due to their thermal motion and disorder. The absolute configuration of **25** was thus established as 8 S, 9 R, 10 S, 13 S, 14 S, 17 S, based on the Flack [0.122 (10)] parameters. Crystallographic data are as follows. Refined formula: C₁₁₈.₃₃H₁₂₅.₀₄Cl₁₂N₂₄O₄.₆₈Zn₆, formula weight (Mr): 2775.94, crystal system: monoclinic, space group: C2, Z = 4, Rint = 0.0634. Lattice parameters, R-factor on F2 > 2σ(F2), and weighted R-factor are as follows: a = 32.8346(8) Å, b = 14.4445(2) Å, c = 30.7569(8) Å, β = 100.966(2)°, V = 14321.0(6) Å³, R = 0.0977, wR = 0.3103, and S = 1.197. The calculated density is 1.287. The linear absorption coefficient (μ) is 0.288. The Flack parameter, determined using 12,777 intensity quotients (Parsons' method), is 0.122(10).

**Reporting summary.** Further information on research design is available in the Nature Research Reporting Summary linked to this article.

## Data availability

The data supporting the findings of this study are available within the article and its Supplementary Information Files, source files or from the corresponding authors on request. The crystallographic data that support the findings of this study are available from the Protein Data Bank (http://www.rcsb.org). The coordinates and the structure factor amplitudes for the apo structures of SptF wild-type apo, SptF wild-type in complex with **1** and α-KG, SptF wild-type in complex with **6** and NOG, and SptF N65T variant in complex with **1** and NOG, SptF S114A variant in complex with **15** and NOG, and SptF Δ9 variant in complex with NOG were deposited under accession code 7EYR[64], 7EYS[65], 7EYT[66], 7EYU[67], 7EYW[68], and 7FCB[69], respectively. The crystallographic information file (CIF) for crystal structure of **25** has been deposited to The Cambridge Crystallographic Data Centre (CCDC), under reference number 2090716. Source data are provided with this paper.

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

## Acknowledgements

We thank Drs. Takayoshi Awakawa and Richiro Ushimaru for critical reading of the manuscript. We also thank Dr. Kunihisa Sugimoto for synchrotron radiation experiments performed at the BL02B1 of the SPring-8. The synchrotron radiation experiments were performed at the BL-1A of the Photon Factory. NMR data were measured at RIKEN Yokohama NMR Facility. We thank Drs. Huiping Zhang and Fumiaki Hayashi for NMR measurements. This work was supported in part by a Grant-in-Aid for Scientific Research from the Ministry of Education, Culture, Sports, Science and Technology, Japan (JSPS KAKENHI Grant Number JP16H06443, JP19K15703, JP20H00490, JP20KK0173, and JP20K22700), the New Energy and Industrial Technology Development Organization (NEDO, Grant Number JPNP20011), AMED (Grant Number JP21ak0101164), the PRESTO program from Japan Science and Technology Agency (JPMJPR20DA), Astellas Foundation for Research on Metabolic Disorders, Takeda Science Foundation, Noda Institute for Scientific Research, and Partnership to Realize Innovative Seeds and Medicines from Sumitomo Dainippon Pharma Co., Ltd. H.T. is a recipient of the JSPS Postdoctoral Fellowship for Foreign Researchers (ID No. P18404).

## Author contributions

T.M. and I.A. designed the experiments. H.T., T.M., H.C., and S.Lyu. performed in vitro analysis and crystallization experiments. A.N. and S.Lee. determined the structure of product using crystalline-sponge method. H.T., T.M., and I.A. analyzed the data. H.T., T.M., and I.A. wrote the paper.

## Competing interests

The authors declare no competing interests.
