## [Peer Review File · Nature Communications]

Molecular insights into the unusually promiscuous and catalytically versatile Fe(II)/ α -ketoglutarate-dependent oxygenase SptFREVIEWER COMMENTS

Reviewer #1 (Remarks to the Author):

This manuscript reported the activity and structure of a multifunctional fungal Fe/2OG enzyme, SptF, involved in terpenoid natural product biosynthesis. The enzyme catalyzes a series of reactions including hydroxylation, epoxidation, desaturation, and rearrangement. The authors also evaluated the substrate promiscuity and discovered that SptF can accommodate substrate analog. In addition, substrate-bound protein structures suggest a flexible substrate-binding lid loop plays a role in dictating the activity.

Prior to further consideration of this manuscript, a few questions might need to be considered and/or addressed.

1. Fe/2OG enzymes are known to catalyze multiple reaction. In this manuscript, other than carbon skeleton rearrangement, other reactivities have been reported and reviewed. Among them, quite a few are reported by the Dr. Abe's research group, e.g. *Nat. Commun.* 2018, 9, 104. There is no doubt that the rearrangement is a fundamentally important and understudied reaction. Although this work demonstrated the involvement of this transformation, from my perspective, without the mechanistic insight and studies, reporting of an Fe/2OG enzyme that can catalyze several reactions is not novel among this enzyme family.

2. In the manuscript, authors carried out the isotope tracer experiments (^{18}O -water and $^{18}\text{O}_2$) to provide insight of the SptF reactions. The summary is reported in Supp. Fig. 3 and Table 1. The authors reported that the substantial ^{18}O from water is incorporated into compound 3, but not compounds 4 and 5. This is the same experimental conditions but carried out at different reaction time periods (10 min vs. 2h). Based on the current mechanism (Figs 1 and 6), 3 is an intermediate for 4 and 5, thus, the influence (^{18}O -water and $^{18}\text{O}_2$) on 3 should be carried through to 4 and 5. In the ^{18}O -water/ $^{18}\text{O}_2$ condition, since a +4 peak was detected in 3, one would expect a +8 signal in 4 and 5 (+6 from the $^{18}\text{O}_2$ and +2 from ^{18}O -water). Observation of no +8 peak in 4 and 5 implies additional steps might be included in the rearrangement. I encourage authors take this lead and look into details. Furthermore, if I am not mistaken, in the reference 41, Bollinger et. al provided spectroscopic and kinetic evidence to draw the conclusion with regard to Fe(IV)=O and Fe(III)-OH ligand exchange with water. In here, without further studies, I am not certain if this conclusion can be made.

Reviewer #2 (Remarks to the Author):

Abe and coworkers have addressed most points satisfyingly and the manuscript has been greatly improved. I only have two small comments:

1. In their response letter, the authors mentioned that ergosterol and lanosterol were tested, but not accepted. This could be mentioned in the manuscript, not only in the response letter.
2. The incubations in ^{18}O labelled water gave small incorporations into 3. Control experiments suggest that this is a non-enzymatic exchange. A possible explanation is addition of water to a carbonyl group to yield a gem-diol. Its collapse back to the carbonyl could proceed with elimination of ^{16}O water, explaining the $^{18}\text{O}/^{16}\text{O}$ exchange. This reaction may happen at C4' of compound 3. This carbonyl carbon becomes less reactive in compound 4, because the 9-OH blocks it, and also less reactive in 5, where it is turned into an ester carbonyl.

Reviewer #3 (Remarks to the Author):

The authors have largely addressed my concerns from the previous review. However, I would like to see the omit maps from the SI shown in a main text figures. All of these maps lack density for certain parts of the substrate component - for the sake of transparency - the actual data should be shown in the main paper so that a reader can easily assess the support for conclusions about the structures.

Reviewer #1:

This manuscript reported the activity and structure of a multifunctional fungal Fe/2OG enzyme, SptF, involved in terpenoid natural product biosynthesis. The enzyme catalyzes a series of reactions including hydroxylation, epoxidation, desaturation, and rearrangement. The authors also evaluated the substrate promiscuity and discovered that SptF can accommodate substrate analog. In addition, substrate-bound protein structures suggest a flexible substrate-binding lid loop plays a role in dictating the activity. Prior to further consideration of this manuscript, a few questions might need to be considered and/or addressed.

1. Fe/2OG enzymes are known to catalyze multiple reaction. In this manuscript, other than carbon skeleton rearrangement, other reactivities have been reported and reviewed. Among them, quite a few are reported by the Dr. Abe's research group, e.g. Nat. Commun. 2018, 9, 104. There is no doubt that the rearrangement is a fundamentally important and understudied reaction. Although this work demonstrated the involvement of this transformation, from my perspective, without the mechanistic insight and studies, reporting of an Fe/2OG enzyme that can catalyze several reactions is not novel among this enzyme family.

We appreciate the reviewer's critical comments. To understand the rearrangement reaction, we investigated in vitro enzyme reactions of the putative intermediates 2, 3, and 7, as the reviewer suggested, in the previous revision. Here in this revision, we further newly performed additional experiment using another putative intermediate 33. This result clearly indicated that 33 is not converted to the final products 3-5, and therefore 33 is not an intermediate in the pathway. We have presented the data in the main text (page 11) and in Supplementary Figure 13. We have also modified the biosynthetic pathway in Figure 7 (previous Figure 6).

Although the reviewer raised concerns about the novelty of this study, SptF exhibits extremely broad substrate specificity toward various meroterpenoids as well as steroids. We would like to emphasize again that this is the first time and very exceptional case. Further, our structure-function analyses of the enzyme successfully provided novel insight into the crucial role of the unique flexible loop region for the substrate recognition and catalysis. We thus believe the present result can stand alone in terms of substance and novelty as a communication of an important discovery.

2. In the manuscript, authors carried out the isotope tracer experiments (^{18}O -water and $^{18}\text{O}_2$) to provide insight of the SptF reactions. The summary is reported in Supp. Fig. 3 and Table 1. The authors reported that the substantial ^{18}O from water is incorporated into compound 3, but not compounds 4 and 5. This is the same experimental conditions but carried out at different reaction time periods (10 min vs. 2h). Based on the current mechanism (Figs 1 and 6), 3 is an intermediate for 4 and 5, thus, the influence (^{18}O -water and $^{18}\text{O}_2$) on 3 should be carried through to 4 and 5. In the ^{18}O -water/ $^{18}\text{O}_2$ condition, since a +4 peak was detected in 3, one would expect a +8 signal in 4 and 5 (+6 from the $^{18}\text{O}_2$ and +2 from ^{18}O -water). Observation of no +8 peak in 4 and 5 implies additional steps might be included in the rearrangement. I encourage authors take this lead and look into details. Furthermore, if I am not mistaken, in the reference 41, Bollinger et. al provided spectroscopic and kinetic evidence to draw the conclusion with regard to Fe(IV)=O and Fe(III)-OH ligand exchange with water. In here, without further studies, I am not certain if this conclusion can be made.

Thank you very much again for the critical comment. According to the suggestion, we carefully repeated the experiments with the same "10 min" reaction time. However, the results were the same as in the previous experiments, only the ratio of water-derived oxygen atoms was slightly decreased. We have presented the results in the main text (page 6-7), supplementary Table 1, and supplementary Figure 3. We believe these results clearly indicated that all oxygen atoms incorporated into 3-5 are mostly derived from O₂.

Regarding the observed differences of ¹⁸O/¹⁶O exchange in 3-5, we have added the following explanation in the main text (page 6) (please also see the response to Reviewer 2).

"A possible explanation is addition of water to a carbonyl group to yield a geminal-diol. Its collapse back to the carbonyl could proceed with elimination of ¹⁶O water, explaining the ¹⁸O/¹⁶O exchange. This reaction may happen at C-4' carbonyl of 3. This carbonyl carbon becomes less reactive in 4, because the C-9 hydroxyl group blocks it, and also less reactive in 5, where it is turned into an ester carbonyl. In fact, the observed ¹⁸O/¹⁶O exchange was not so significant in 4 and 5 as it was in 3."

Finally, in the reference 41, Dr. Bollinger's group used D-labeled substrates for spectroscopic and kinetic analyses to show the Fe(IV)=O and Fe(III)-OH ligand exchange with water. In contrast, in our case, it is quite difficult to prepare site-specific D-labeled substrates due to the complex biosynthetic pathway and highly oxidized chemical structures. Therefore, we have deleted the sentence about the slow exchange of ferryl and/or ferric species with water from the text. We expect to fulfill these studies by future collaborations with experts in chemical synthesis and metabolic engineering.

Reviewer #2:

Abe and coworkers have addressed most points satisfyingly and the manuscript has been greatly improved.

We would like to thank the reviewer for the positive comment.

I only have two small comments:

1. In their response letter, the authors mentioned that ergosterol and lanosterol were tested, but not accepted. This could be mentioned in the manuscript, not only in the response letter.

According to the suggestion, we have presented the results in the main text (page 8).

2. The incubations in 18O labelled water gave small incorporations into 3. Control experiments suggest that this is a non-enzymatic exchange. A possible explanation is addition of water to a carbonyl group to yield a gem-diol. Its collapse back to the carbonyl could proceed with elimination of ¹⁶O water, explaining the ¹⁸O/¹⁶O exchange. This reaction may happen at C4' of compound 3. This carbonyl carbon becomes less reactive in compound 4, because the 9-OH blocks it, and also less reactive in 5, where it is turned into an ester carbonyl.

We really appreciate the reviewer's thoughtful and helpful comments. We completely agree with the reviewer and have added the explanation in the main text (page 6) (please also see the response to Reviewer 2).

Reviewer #3:

The authors have largely addressed my concerns from the previous review. However, I would like to see the omit maps from the SI shown in a main text figures. All of these maps lack density for certain parts of the substrate component - for the sake of transparency - the actual data should be shown in the main paper so that a reader can easily assess the support for conclusions about the structures.

We appreciate the reviewer's positive comment. According to the suggestion, we have included the omit maps (previous supplementary figure 8) as a new Figure 5 in the main text.

REVIEWERS' COMMENTS

Reviewer #1 (Remarks to the Author):

The authors have addressed the questions that I raised in the earlier review. In general, the changes and additions they have made the manuscript much stronger.

Reviewer #2 (Remarks to the Author):

The authors have addressed all comments satisfyingly. The manuscript can now be accepted for Nat. Commun.

Reviewer #1:

The authors have addressed the questions that I raised in the earlier review. In general, the changes and additions they have made the manuscript much stronger.

We appreciate the reviewer's comment.

Reviewer #2:

The authors have addressed all comments satisfyingly. The manuscript can now be accepted for Nat. Commun.

We appreciate the reviewer's comment.